# Numerical Analysis of Dynamic Properties of an Auxetic Structure with Rotating Squares with Holes

**DOI:** 10.3390/ma15248712

**Published:** 2022-12-07

**Authors:** Agata Mrozek, Tomasz Strek

**Affiliations:** Institute of Applied Mechanics, Poznan University of Technology, Jana Pawla II 24, 60-965 Poznan, Poland

**Keywords:** auxetic, effective properties, dynamics, vibration transmission loss, transmissibility, mechanical impedance, rotating units

## Abstract

In this paper, a novel auxetic structure with rotating squares with holes is investigated. The unit cell of the structure consists of four units in the shape of a square with cut corners and holes. Finally, the structure represents a kind of modified auxetic structure made of rotating squares with holes or sheets of material with regularly arranged diamond and square cuts. Effective and dynamic properties of these structures depend on geometrical properties of the structure. The structures are characterized by an effective Poisson’s ratio from negative to positive values (from about minus one to about plus one). Numerical analysis is made for different geometrical features of the unit cells. The simulations enabled the determination of the dynamic characteristic of the analyzed structures using vibration transmission loss, transmissibility, and mechanical impedance. Numerical calculations were conducted using the finite element method. In the analyzed cases of cellular auxetic structures, a linear elasticity model of the material is assumed. The dynamic characteristic of modified rotating square structures is strongly dependent not only on frequency. The dynamic behavior could also be enhanced by adjusting the geometric parameter of the structure. Auxetic and non-auxetic structures show different static and dynamic properties. The dynamic properties of the analyzed structures were examined in order to determine the frequency ranges of dynamic loads for which the values of mechanical impedance and transmissibility are appropriate.

## 1. Introduction

Composite materials can have different physical properties combined. The interest in this scientific issue is due to the possibility of applying the results obtained in practice. Due to these facts, some terminology and neologisms have been proposed and collected by Lakes in the book [1] concerning the properties of heterogeneous materials: smart materials (the properties can be altered or controlled by field variable stimuli); bio-mimetic (the design imitates structures observed in biological materials); bio-inspired (the design process is influenced by biological materials); auxetic materials (have a negative Poisson’s ratio); metamaterials (allow unusual properties); architected materials (multiphase materials or cellular materials that are controlled and optimized to achieve specific functions or properties); multifunctional materials (serve multiple roles in a practical application).

Saint-Venant (1848) seems to be the first who suggested that Poisson’s ratio might be negative in anisotropic solids and achieve values greater than 1/2 [2,3]. The histories of Poisson’s ratio (PR) and negative Poisson’s ratio (NPR) are presented in the first chapter of Lim’s book [2]. Materials that become wider when they are stretched and become thinner when compressed are characterized by a negative Poisson’s ratio and are called auxetic materials (auxetics).

Auxetics are examples of metamaterials with negative properties [4]. The terms “auxetic” and “auxetic materials” were introduced by Evans et al. in 1991 [5]. The Greek word “auxeticos” is defined as “that which tends to increase”.

A number of review papers and books have been written concerning auxetic materials. These include the books by Lim, Lakes [1,2,6], and the works of many researchers [7,8,9,10,11,12,13,14]. Nevertheless, they are not limited to them.

The range of research topics on auxetic materials increases exponentially, resulting in a large number of publications over the past decade. Hence, interested readers are referred, but not limited, to papers in the area of auxetics written by Wojciechowski et al. [15,16], Novak et al. [17,18,19], Lim [20], Strek et al. [21,22,23], Shepard et al. [24], Jopek et al. [25,26], Michalski et al. [27,28,29], Burlaga et al. [30], and Zhang et al. [31].

Due to the mechanism of displacements and deformations in the auxetic structure, we divide them into the following types: reentrant type; chiral type; and rotating units. In particular, Saxena et al. [9] summarized different types of auxetic structures and models as

(1)re-entrant type (2D re-entrant, 2D re-entrant triangular, 3D re-entrant, double arrowhead);(2)rotating polygons (squares, rectangles, rhombi, parallelograms, triangles, tetrahedral);(3)chiral type (chiral circular, rotachiral, 3D chiral, anti-chiral);(4)perforated sheets (perpendicularly oriented cuts, randomly oriented cuts, diamond perforations, star perforations, 2D sheet containing holes);(5)crumpled sheets (aluminum thin foils, graphene sheet);(6)other (nodule fibril model, hexatruss, egg rack structure, missing rib, generalized tethered nodule, entangled single wire auxetic, grooved block of metal, hard discs).

Teik-Cheng Lim presented in his works [6,32] the identification of structures due to the mechanical model of operation. This analysis identified five categories of auxetic models. The models of the auxetic structure in each category are similar to each other. The author distinguished the following categories of structures in which unit cells are characterized by (1) single periodicity, (2) double periodicity, (3) synchronized direction of rotation of their rotating units, (4) random rotation and (5) no rotation.

Auxetic structures with rotating units were presented by Grima and Evans [33], where rigid rectangular and triangular cells were connected together at selected vertices by hinges. The later papers on this topic of auxetics made of rotating units were published by researchers from Malta [14,34,35,36,37] and Slovenia [38]. Grima-Cornish et al. [14] described the role of rotating rigid units in generating negative thermal expansion (NTE) and negative compressibility (NC) in auxetic materials with a mechanism made of rotating units (RUM) or quasi-RUM (qRUM). Han et al. [39] investigated the influence of the hole radius in the design domain of an auxetic structure with rotating units on the mechanical properties of the structure and the importance of the materials in different parts of the design domain to the overall stability of the structure.

There are a lot of properties important to auxetic materials. These properties include indentation resistance, synclastic, shear properties, fracture toughness, first law of thermodynamics for auxetics, wave propagation/vibration transmission, acoustic absorption, variable permeability concomitantly with a high shear modulus relative to the bulk modulus, negative compressibility [2,6,8,40,41].

Specific applications of auxetic materials and structures require a lot of further development. Nevertheless, a review of the literature already shows the great potential for their application in various fields. One can mention here, for example, the following applications of auxetics: medical and biomedical applications (e.g., stents, dilators for opening the cavity of an artery, coronary angioplasty and related procedures, smart bandages, smart filters); sports applications (sport helmets, protection against extreme weather during outdoor/adventure sports), molecular sieves and filters and defouling applications; displacement amplifiers; auxetic cushions in seats; auxetic springs; acoustic-to-electrical energy conversion; electromagnetic applications in antennas and reflectors; applications in fabrics, garments, and fashion apparel; fabrication of 3D auxetic fabric; auxetic shape memory alloys; footwear and articles of apparel; etc. One can find more details about applications of auxetic materials and structures in previous articles [7,8,9,41,42,43].

The study of vibrations is important from the point of view of the analysis of materials, structures [44,45,46], continuous systems with constraints, and controlled systems or oscillators [47,48]. The vibrations imply cyclical stresses, so fatigue failure is inevitable. Teik-Cheng Lim presented and discussed advanced issues concerning vibration damping, damping of acoustic vibration, and auxetic solids and structures [2]. Other authors and researchers also investigated a lot of properties of auxetic materials and structures (better vibration or acoustic properties compared to classic materials with positive Poisson’s ratios, impact energy absorption).

Their dynamic behavior has not been widely studied. In the study [44], the influence of two features of the lattice structure (the strut diameter and the mass attached to the sample) on the obtained frequency response function was evaluated. The effect of geometrical parameters on the dynamic response of the structure was also analyzed in the studies [45,46]. It was proved that the structures with negative Poisson’s ratios could achieve a wider bandwidth and a lower frequency bandgap than structures with positive Poisson’s ratios. According to Tao et al. [46], the bandwidth increases about 15.1% compared to a structure with a positive Poisson’s ratio. The parametric study was also presented in the paper [45]. It was proved that the adjustment of geometric features of a novel lightweight bidirectional re-entrant lattice metamaterial enabled changing the broad bandgaps in the low-frequency range. Strek et al. analyzed the dynamic response of a three-layered sandwich beam with a metal foam core. The study showed that the Poisson’s ratio of the foam core influences the obtained value of the average mechanical impedance of the sandwich beam. It observed that the lower the values of Poisson’s ratios, the higher the values of the resonance frequencies [23]. 

Since 1987, when Lakes manufactured the first auxetic re-entrant foam structure, we have observed an increase in research in auxetic materials and structures. However, there are still many important limitations and properties that must be studied before their real-life implementations. The challenges to be solved include the static and dynamic properties of auxetics. Extensive studies have been conducted analyzing rotating rigid units and their role in generating auxetic behavior in different classes of materials, such as foams, silicates and zeolites. The potential applications of auxetics include, but are not limited to, biomedical devices (e.g., novel stent design [49,50,51]), sport applications [11], military and industrial devices (e.g., explosion protection and blast response structures [29], energy absorption structures [17], sandwich panels [52]). In this paper, the dynamic properties of auxetic and non-auxetic structures are investigated. Structures with effective PRs from negative to positive values are investigated. The effective PR value depends on the geometrical properties of the structure and does not depend on the component material. The structures represent kinds of modified auxetic structures made of rotating squares with holes or sheets of material with regularly arranged diamond and square cuts. 

Numerical analysis was made for different geometrical features of the unit cells. The simulations enabled the determination of dynamic characteristics of the analyzed structures using mechanical impedance (*Z*), transmissibility (vibration transmission, *T*), and vibration transmission loss (*VTL*). The basic dynamic properties of the analyzed structure were examined in order to determine the frequency ranges of dynamic loads for which the values of mechanical impedance and transmissibility were appropriate. In the case of better/greater damping of vibration, these are frequency ranges for which *T* takes negative values. Whereas, when we want to give the structure with a lower force the set vibration velocities, these are the frequency ranges for which *Z* is small.

Numerical calculations were conducted using the finite element method. In the analyzed cases of cellular auxetic structures, it is assumed that a linear elasticity model of material is used in cellular structure. The unit cells of the structures analyzed in this paper can be used as the main build element of bigger and more complex structures such as stents or cores of sandwich panels. Although in this paper 2D plane stress models of 3D solid structures are studied, the full 3D structures with rotating squares with holes can be used in real-life applications. 

Moreover, the presented method of analysis of static and dynamic mechanical parameters of structures can be used in the analysis of 3D models of solids and structures. Based on the results of testing the static (effective Poisson’s ratio, Young’s modulus) and dynamic (mechanical impedance, vibration transmissibility) mechanical properties of representative samples of cellular materials (unit cells), we can use them in the construction of more complex materials and structures. An example of such constructions are sandwich panels, the core of which may be materials made of periodic unit cells, which were analyzed in this work. The results of the work carried out so far indicate very wide possibilities for their use in mechanical engineering, biomedical, automotive, military and building constructions. 

## 2. Mathematical Model-Governing Equation

### 2.1. Linear Elasticity Equations

This section considers the equations which govern the motion of elastic structures experiencing small deformations, the determination of effective mechanical properties (Poisson’s ratio, Young’s modulus), and dynamic properties (mechanical impedance, vibration transmission loss or transmissibility).

The Navier-Cauchy equation of motion describes elastodynamics, and we can write this equation in displacement formulation as: (1)$$\rho \frac{\partial^{2}u}{\partial t^{2}}-\nabla \cdot S=F_{V}$$
where *ρ* is the density, **u** is the vector of displacements, $u=\left[u,v,w\right]=\left[u_{1},u_{2},u_{3}\right]$, S is the stress tensor, $F_{V}$ is the volume force vector.

We can represent the balance of forces in three directions in the following form: (2)$$-\nabla \cdot S=F_{V}$$

The total stress S in Hooke’s law (constitutive equation, relation between stresses and deformations) is: (3)S=C:ε
and the elastic strain tensor ε is: (4)$$\varepsilon =\frac{1}{2}\left({\left(\nabla u\right)}^{T}+\nabla u\right)$$

In the case of the isotropic linear material, the elasticity matrix becomes as follows:(5)$$C=\frac{E}{\left(1+\nu \right)\left(1-2\nu \right)}\left[\begin{array}{ccc} \begin{array}{cc} 1-\nu & \nu \\ \nu & 1-\nu \end{array} & \begin{array}{cc} \nu & 0\\ \nu & 0 \end{array} & \begin{array}{cc} 0 & 0\\ 0 & 0 \end{array}\\ \begin{array}{cc} \nu & \nu \\ 0 & 0 \end{array} & \begin{array}{cc} 1-\nu & 0\\ 0 & \frac{1-2\nu }{2} \end{array} & \begin{array}{cc} 0 & 0\\ 0 & 0 \end{array}\\ \begin{array}{cc} 0 & 0\\ 0 & 0 \end{array} & \begin{array}{cc} 0 & 0\\ 0 & 0 \end{array} & \begin{array}{cc} \frac{1-2\nu }{2} & 0\\ 0 & \frac{1-2\nu }{2} \end{array} \end{array}\right]$$

In terms of Lamé parameters λ and μ, the elasticity matrix can be written as:(6)$$C=\left[\begin{array}{ccc} \begin{array}{cc} \lambda +2\mu & \lambda \\ \lambda & \lambda +2\mu \end{array} & \begin{array}{cc} \lambda & 0\\ \lambda & 0 \end{array} & \begin{array}{cc} 0 & 0\\ 0 & 0 \end{array}\\ \begin{array}{cc} \lambda & \lambda \\ 0 & 0 \end{array} & \begin{array}{cc} \lambda +2\mu & 0\\ 0 & \mu \end{array} & \begin{array}{cc} 0 & 0\\ 0 & 0 \end{array}\\ \begin{array}{cc} 0 & 0\\ 0 & 0 \end{array} & \begin{array}{cc} 0 & 0\\ 0 & 0 \end{array} & \begin{array}{cc} \mu & 0\\ 0 & \mu \end{array} \end{array}\right]$$

The constitutive equation for a linear elastic isotropic material can be described as:(7)S=2μ ε+λ(∇·u)I
where I is the identity matrix.

In this case, Navier’s equation of motion for the linear elastic isotropic material and with zero volume force ($F_{V}=0$) can be written as: (8)$$\rho \frac{\partial^{2}u}{\partial t^{2}}-\left(\mu \nabla^{2}u+\left(\lambda +\mu \right)\nabla \nabla \cdot u\right)=0.\;$$

According to the harmonic displacement of the solid, the equation of harmonic motion with the assumed linear-elastic model of the material meets the formula:(9)$$-\rho \omega^{2}u-\left(\mu \nabla^{2}u+\left(\lambda +\mu \right)\nabla \nabla \cdot u\right)=0,$$
where *µ*, *λ* are Lamé constants.

The loss factor ($\eta_{S}$) is the ratio of the average power loss to the peak load loss over the time interval that the load acts on the system [53]:(10)$$C_{d}=\left(1+i\;\eta_{S}\right)C$$

In the case of analyzing two-dimensional plane problems, the basic equations of elasticity can be simplified due to the made assumptions (some strain or stress components are by definition equal to zero), in relation to the equations describing three-dimensional spatial problems.

There are two main 2D models of plane solid mechanics problems [54]:plane stress—the load acts in a plane, and the thickness of the structure is constant and appropriately small in relation to other dimensions of the structure (appropriate thinness);plane strain—a constant load acts along the entire length of the structure (large enough), and its cross-section is constant over the entire length of its thickness.

For the plane stress, all stress components in the third direction (normal and shear) perpendicular to x–y plane are assumed zero. In this case, the elasticity matrix becomes:(11)$$C=\frac{E}{1-\nu^{2}}\left[\begin{array}{ccc} 1 & \nu & 0\\ \nu & 1 & 0\\ 0 & 0 & \frac{1-\nu }{2} \end{array}\right]$$

Substitution of the elastic matrix for plane stress in the equilibrium equation gives Navier’s equation in terms of displacements as follows:(12)$$G\nabla^{2}u+\frac{E}{2\left(1-\nu \right)}\frac{\partial }{\partial x}\left(\frac{\partial u}{\partial x}+\frac{\partial v}{\partial y}\right)=0,\;G\nabla^{2}v+\frac{E}{2\left(1-\nu \right)}\frac{\partial }{\partial y}\left(\frac{\partial u}{\partial x}+\frac{\partial v}{\partial y}\right)=0.$$
where G=μ=E/(2(1+ν)) is the shear modulus. The plane stress model of solid mechanics is used in this paper.

### 2.2. Effective Mechanical Properties of the Structure

The effective Poisson’s ratio can be calculated using the following formula: (13)$$PR_{eff\_ij}=-\frac{\langle \varepsilon_{j}\rangle }{\langle \varepsilon_{i}\rangle },$$
where $\langle \varepsilon_{j}\rangle$ and $\langle \varepsilon_{i}\rangle$ are average strains in a transverse and longitudinal direction, respectively. Longitudinal direction is the direction of the applied load. The average strains can be calculated using:(14)$$\langle \varepsilon_{i}\rangle =\frac{\langle dL_{i}\rangle }{L_{i}},$$
where $L_{i}$ and $\langle dL_{i}\rangle$ are the original dimension of the sample and the average change of the length in *i*-th direction, respectively.

The effective Young’s modulus can be calculated using the following formula [55]:(15)$$E_{eff\_i}=\left|\frac{\langle \sigma_{i}\rangle }{\langle \varepsilon_{i}\rangle }\right|,$$
where $\langle \sigma_{i}\rangle =\frac{F_{i}}{\langle A_{i}\rangle }$ is the average strain, $\langle A_{i}\rangle =\frac{V_{}}{L_{i}}$ is the average cross-section area in *i*-th direction, $F_{i}$ is a force applied in *i*-th direction, and $V_{}=th\int_{A}dA$ is the wrapping volume of the sample when the thickness of 2D sample is *th*.

The average dimensions of the sample in *x* and *y* directions can be calculated using:(16)$$\langle dL_{x}\rangle =\langle dL_{1}\rangle =\langle u_{1}(\mathrm{RB})\rangle -\langle u_{1}\left(\mathrm{LB}\right)\rangle ,$$
(17)$$\langle dL_{y}\rangle =\langle dL_{2}\rangle =\langle u_{2}\left(\mathrm{TB}\right)\rangle -\langle u_{2}\left(\mathrm{BB}\right)\rangle ,$$
respectively.

The values $\langle u_{1}(RB)\rangle$ and $\langle u_{1}(LB)\rangle$ are average values of the second component ($u_{1}$) of displacement vector (u) on the right (RB) and left (LB) boundaries of the sample:(18)$$\langle u_{i}(\mathrm{RB})\rangle =\frac{\int_{\mathrm{RB}}^{}u_{i}\;dx_{i}}{\int_{\mathrm{RB}}^{}dx_{i}},\;\langle u_{i}(\mathrm{LB})\rangle =\frac{\int_{\mathrm{LB}}^{}u_{i}\;dx_{i}}{\int_{\mathrm{LB}}^{}dx_{i}}.$$

The values $\langle u_{2}(\mathrm{TB})\rangle$ and $\langle u_{2}(\mathrm{BB})\rangle$ are average values of the second component ($u_{2}$) of displacement vector (u) on the top (TB) and bottom (BB) boundaries of the sample:(19)$$\langle u_{i}(\mathrm{TB})\rangle =\frac{\int_{\mathrm{TB}}^{}u_{i}\;dx_{i}}{\int_{\mathrm{TB}}^{}dx_{i}},\;\langle u_{i}(\mathrm{BB})\rangle =\frac{\int_{\mathrm{BB}}^{}u_{i}\;dx_{i}}{\int_{\mathrm{BB}}^{}dx_{i}}$$

### 2.3. Dynamic Properties of the Structure

The dynamic reaction of the structure to the mechanical vibrations acting on it can be divided into two categories. The first category, “to the structure”, defines the dependence of the motion of the structure as a function of frequency, expressed as a function of the mechanical impedance of the driving point. The second category, “through structure” response function, is defined as the vibration transmission from the boundary with the applied load to the free opposite boundary of the structure.

The frequency domain analysis enabled the computation of the response of a model subjected to harmonic excitation for the specified range of frequencies. The dynamic characteristic was defined using mechanical impedance (*Z*), vibration transmission loss (*VTL*), and transmissibility (*T*). Mechanical impedance (*Z*) is defined as the ratio of applied force (*F*_0_) and the velocity response (*v*_0_), which strongly depends on frequency [56,57]. In order to evaluate the dynamic properties of the whole domain (*Ω*), the average mechanical impedance was analyzed. It is expressed using the following formula [23]:(20)$$\langle Z\rangle =\frac{\Omega \;F_{0}}{\int_{\Omega }v_{0}d\Omega }.$$

In this study, transmissibility (*T*) is evaluated in terms of displacements. The displacement transmissibility, expressed in dB, is defined as the ratio between the amplitude of the response and the excitation in the respective direction [58]. It is evaluated using the given formula: (21)$$T_{i}=10\log_{10}\left(\frac{u\_r_{i}}{u\_e_{i}}\right),$$
where $u\_r_{i}$ indicates the amplitude of response and $u\_e_{i}$ is the amplitude of the harmonic excitation in the indicated *i*-*th* direction (i=x,y,z). 

Another essential parameter which enables determination of the dynamic characteristic of analyzed structures is vibration transmission loss (*VTL*). This parameter allows the evaluation of the isolation properties of the structures. It indicates the frequency range in which the structure does not transmit vibration [59,60]. It is expressed using the following formula:(22)$$VTL=10\;\log_{10}\left(\frac{u\_e_{i}}{u\_r_{i}}\right).$$

As is well-noted, the vibration transmission loss is the reverse of transmissibility. 

## 3. Models and Methods

### 3.1. Geometrical Model

The analyzed structure consists of a unit cell which is based on rotating squares. The single unit cell consists of four squares with holes of different sizes. The structure is created by cutting four triangles from the surface of a single square. 

In the model, the four geometric parameters are determined. All parameters are defined as the ratio of the variable dimension to the basic square edge length ($L_{x}$). The first dimension (*d*_1_) is defined as one of the edges of the cut-out rectangle (Figure 1). The second parameter (*d*_2_) is calculated based on the ratio d_1_. It was assumed that the area of cut-out triangles is constant (Figure 2). The parameter *d*_1_ was expressed as the ratio between the distance *l*_1_ and the length of the square edge. It was studied in the range of 0.30–0.65 (Figure 3 and Figure 4). 

The next essential geometric feature of the structure is the distance between the outer square and the inner square (*w*_1_). The parametrized dimension is shown in Figure 5. The inner square is cut from the surface of the base shape. The *w*_1_ parameter was analyzed in the range of 0.15–0.45 (Figure 6 and Figure 7). 

The last analyzed parameter was the radius of the fillet in the middle part of a unit cell. The parameter has a value from 0 to 0.15 (Figure 8, Figure 9 and Figure 10).

All analyzed structures have dimensions: $L_{x\;}\times L_{y}$, where $L_{x\;}=L_{y\;}=0.1$ m. 

Designation and definition of boundaries of analyzed samples:right boundary (RB): $x=L_{x}\;$ and $y\in \langle 0,L_{y}\rangle$;left boundary (LB): x=0 and $y\in \langle 0,L_{y}\rangle$;bottom boundary (BB): y=0 and $x\in \langle 0,L_{x}\rangle$;top boundary (TB): $y=L_{y}$ and $x\in \langle 0,L_{x}\rangle$.


### 3.2. Finite Element Model

During the conducted analysis, two load cases were considered. In the first load scenario, in order to evaluate the effective value of Poisson’s ratio and Young’s modulus, on the edges indicated in Figure 6, roller constraint boundary conditions were applied, which is equivalent to a condition of symmetry. It resulted in zero displacements in the perpendicular direction to the indicated edge. Nevertheless, the boundary can move in a tangential direction. In the second case, the unit cells were harmonically loaded on the top or bottom boundaries. 

Boundary conditions (BC), see Figure 11 and Figure 12, were applied to analyze the mechanical properties of structures when the load applied at the upper (top) boundary is directed vertically in the y-direction: boundary (RB): free BC; boundary (LB): roller (symmetry) BC: n·u=0, where n is the normal unit vector to the boundary;boundary (BB): roller (symmetry) BC: n·u=0, boundary (TB): prescribed displacement: $u_{0}=\left(0,\Delta L_{y}\right)$; orboundary (TB): applied force: $S\cdot n=F_{A}$, $where\;F_{A}=\left[0,\frac{F_{2}}{A}\right]$ and A=th·|TB|, where |TB| is the length of boundary TB.


Boundary conditions (BC), see Figure 13, were applied to analyze the dynamic properties of structures when the load applied at the bottom boundary is directed vertically in the y-direction: boundary (RB): free BC;boundary (LB): free BC;boundary (TB): free BC;boundary (BB): harmonic excitation prescribed displacement BC: $u_{0}=\left(0,u\_e_{i}\right)$; orboundary (BB): harmonic excitation force BC:$\;S\cdot n=F_{A}e^{i\phi }$, $where\;F_{A}=\left[0,\frac{F_{2}}{A}\right]$ and A=th·|BB|. 


The influence of geometrical features was investigated using the finite elements method (FEM) [61]. The analyses were conducted in Comsol Multiphysics software with the MUMPS solver [53]. The geometry was discretized using triangular elements. Due to the computational cost, the two-dimensional model was applied. The 2D physics interface for plane stress assumes that the load is constant throughout the material’s thickness (*th*). Second order Lagrangian polynomials were used as shape functions for finite triangle elements. Contact without friction between structure elements was assumed.

Convergence of the simulation was tested both for coarser and finer mesh, which consists of almost 9350 elements. The obtained relative differences between results were below 5%.

## 4. Numerical Results

### 4.1. Effective Mechanical Properties of the Analyzed Structure

The numerical experiments were conducted in order to analyze the mechanical properties of the proposed auxetics structure. The influence of geometric features on the Poisson’s ratio and Young’s modulus was evaluated. All numerical calculations were conducted using the finite element method and a plane stress solid mechanics model with a linear elasticity model of the material.

The applied material is structural steel (Table 1). 

When calculating the value of PR_eff_, $\Delta L_{y}$ was defined as the prescribed compression displacement in the y-direction applied on the top boundary. The value of the prescribed displacement was evaluated using the following formula:(23)$$\Delta L_{y}=\frac{L_{y}}{20}.\;$$

The value of the applied force in every conducted simulation (with the exception of analyses that allowed the determination of PR_eff_) was equal to 100 N, and the force acted parallel to the *y*-axis.

The area of each cut triangle (see Figure 1) was constantly equal to $P_{0}=\left(d_{1}d_{2}\right)/2=0$.1 for each parameter $d_{1}$. Real dimensions represented by the parameters can be calculated using formula: (24)$$l_{1}=d_{1}\cdot L_{x}/2,\;$$
where $L_{x}$ is assumed to equal to 0.1 m. The thickness of the structure was equal to *th* = 0.01 m. The range of values of the analyzed parameters is shown in Table 2. 

The Poisson’s ratio (υ) is defined as the ratio of the negative transverse strain and the axial strain in the direction of the applied force. In this study, the obtained value of the effective Poisson’s ratio (Equation (13)) was in the range of −1 to 1 (Figure 14). It was noticed that the value of υ_eff_ strongly depends on the geometric features of the structures. It was observed that the higher the value of *w*_1_, the lower the value of υ_eff_, and the structure was more auxetic. For *w*_1_ values equal to 0.15 in all of the analyzed cases, the structures were non-auxetic. For this instance, the value of υ_eff_ was less influenced by changes in the *d*_1_ parameter. It was also observed that the higher the value of *w*_1_, the wider the range of *d*_1_ in which the structure is auxetic. The maximum υ_eff_ occurred for a *d*_1_ equal to about 0.45.

For the structure in which no auxetic effect was observed (*w*_1_ = 0.15), a significant decrease in the value of the effective Young’s modulus (Equation (15)) was observed (*E*_eff_). The ratio *E*_ff_/*E*_s_ was no higher than 0.01 (Figure 15). The slight fluctuations of the values with the change of the *d*_1_ parameter were noticed. In the case of structures which obtained negative Poisson’s ratios, properties of the structure changed significantly with the increase in the value of *d*_1_. For the case in which υ_eff_ reached a value approximately equal to 0 (*w*_1_ = 0.45, *d*_1_ = 0.45), the ratio *E*_ff_/*E*_s_ reached the maximum. *E*_eff_ for this parameter set was about 0.46 times lower than the value of the material’s Young’s modulus. The d_1_ variable also influenced the obtained structure’s properties. It was noticed that the wider the range of *d*_1_ in which the structure is non-auxetic, the smaller the changes in the value of E_eff_.

Another geometric feature that influenced the obtained effective properties of the structure was the value of the radius of the fillets (Figure 16, Figure 17 and Figure 18). For the range of *d*_1_ parameters in which the structures were auxetic, the fillet had a significant impact on the resulting value of υ_eff_. The larger the radius, the less auxetic the structure. In all analyzed cases, it was observed that the higher the value of υ_eff_, the smaller the influence of the fillet radius. 

### 4.2. Mechanical Impedance

In this research, a study of the influence of geometric features of the analyzed structure on the dynamic characteristic was conducted. The frequency of the applied harmonic load changes in the range of 50–20,200 Hz (step 1 Hz) for mechanical impedance analysis and in the range of 50–5050 Hz (step 0.5 Hz) for the remaining cases. 

Mechanical impedance (Equation (20)) is defined as the frequency-dependent relationship between the applied harmonic load and the velocity of an object. The analysis of mechanical impedance was conducted in the frequency range of 1 Hz to 20,200 Hz (Figure 19 and Figure 20). 

The lower the value of mechanical impedance, the lower the force magnitude applied in order to cause motion at a specific velocity. The obtained value of mechanical impedance strongly depends on the frequency. The changes in the geometric parameters of the structure cause significant changes in mechanical impedance values. In the frequency range of 1–20,200 Hz, the dynamic behavior strongly depends on the parametric features of the structure, which makes them auxetic or non-auxetic. The frequencies in which the minimum value of *Z* was achieved changed, as well as the number of local minimums of the function. 

One of the obtained auxetic structures (*w*_1_ = 0.45 and *d*_1_ = 0.3) had a wider frequency gap band than other structures for frequencies in the range 1–10,000 Hz, as shown in Figure 11. Another auxetic structure (*w*_1_ = 0.35, *d*_1_ = 0.3), for the frequency above 17,000 Hz, demonstrated the most advantageous influence on the obtained dynamic characteristic. The value of mechanical impedance fluctuated in the range of 10^2^ kg/s to 10^4^ kg/s. For the whole analyzed range of *d*_1_ parameter, the larger the radius of the fillet, the higher the frequency value in which the local minimum is observed (Figure 21). 

### 4.3. Vibration Transmission Loss (VTL)

Vibration transmission loss (Equation (22)) of the structures was investigated using the finite element method. The vibration transmission loss of auxetic structures reaches a higher value than structures with a positive value of the effective Poisson’s ratio (Figure 22 and Figure 23). The lowest value of *VTL* was achieved in a non-auxetic structure for which the variable *d*_1_ was 0.45 (for both cases of *w*_1_). With a higher value of *w*_1_, the obtained *VTL* reached a more beneficial level.

### 4.4. Transmissibility

Displacement transmissibility was applied in order to assess the vibration transmission at different frequencies. This characteristic is widely used in the area of vibration isolator design. Transmissibility (Equation (21)) was studied in the range of 1–5050 Hz. The least beneficial effect on transmissibility was observed for non-auxetic structures (*w*_1_ = 0.45, *d*_1_ = 0.45), as shown in Figure 24 and Figure 25. In the case of auxetic structures, the applied displacement was significantly extinguished.

## 5. Conclusions

The influence of geometric features on the obtained dynamic characteristic of the structure was evaluated. The dynamic properties of the structure were determined using parameters such as mechanical impedance (*Z*), vibration transmission loss (*VTL*), and transmissibility (*T*). The effective properties of an auxetic structure with rotating squares with holes were calculated as functions of geometric parameters of structures, as well.

The conducted analysis showed that the geometric parameters of the unit cell significantly affect the obtained effective properties of the structure. The achieved value of υ_eff_ for various parameter sets was in the range of −1 to 1. The smaller the dimension of the cut-out inner square, the lower the value of υ_eff_ observed. This parameter strongly whether the structure is auxetic. The dimensions of the edges of the cut-out rectangle (*d*_1_ and *d*_2_) cause changes in the value of υ_eff_ in the range of 0.43 (for *w*_1_ = 0.15) to 1.61 (for *w*_1_ = 0.3). The geometric features also significantly influence the value of E_eff_.

Another analyzed parameter was the fillet radius. For non-auxetic structures, the fillet radius is not considered a significant parameter affecting the obtained value of υ_eff_. In the case of an auxetic structure, the larger the radius, the higher the value of υ_eff_ observed.

Mechanical impedance defines the relation between the driving force at the input to a body and the resultant velocity of the body. The value of mechanical impedance is strongly dependent on frequency. The dynamic behavior could be enhanced by adjusting the geometric parameter of the structure. Auxetic structures demonstrate a higher value of vibration transmission loss than non-auxetic unit cells. The usage of structures with a negative value of Poisson’s ratio causes the applied displacement to be significantly extinguished.

The dynamic characteristic of modified rotating square structures is strongly dependent not only on frequency. It is also influenced by the applied geometric features of the structure. Adjusting the appropriate set of geometric parameters to the selected application could significantly improve the efficiency of the used structure. 

## Figures and Tables

**Figure 1 materials-15-08712-f001:**
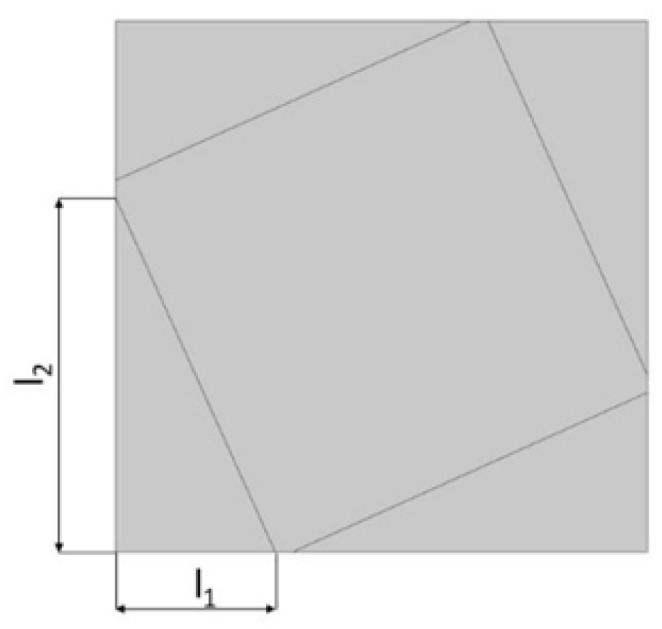
The first stage of the process of obtaining the cell: defining the dimensions of the cut-out triangle.

**Figure 2 materials-15-08712-f002:**
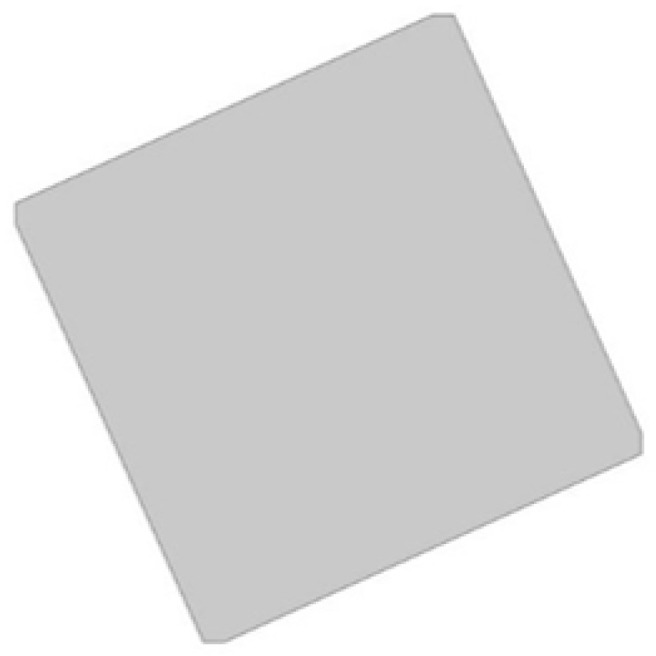
The results of performed process of obtaining the cell.

**Figure 3 materials-15-08712-f003:**
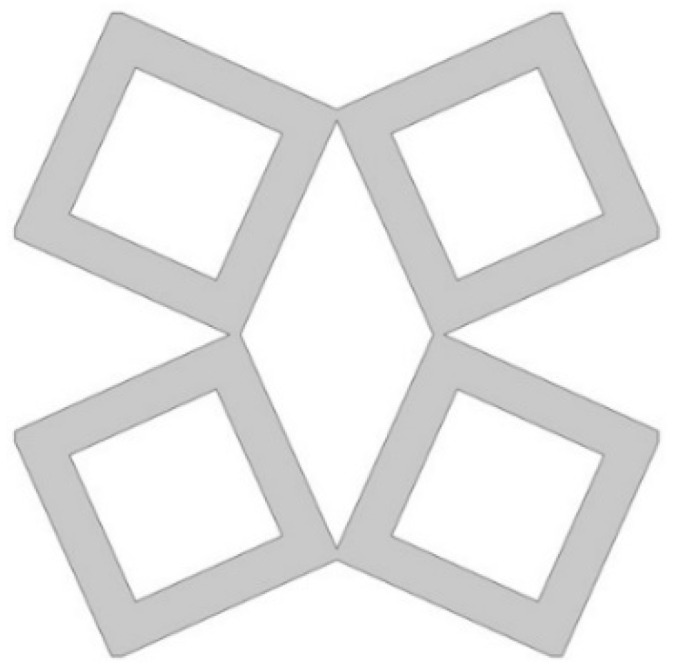
The obtained unit cell for *d*_1_ = 0.30.

**Figure 4 materials-15-08712-f004:**
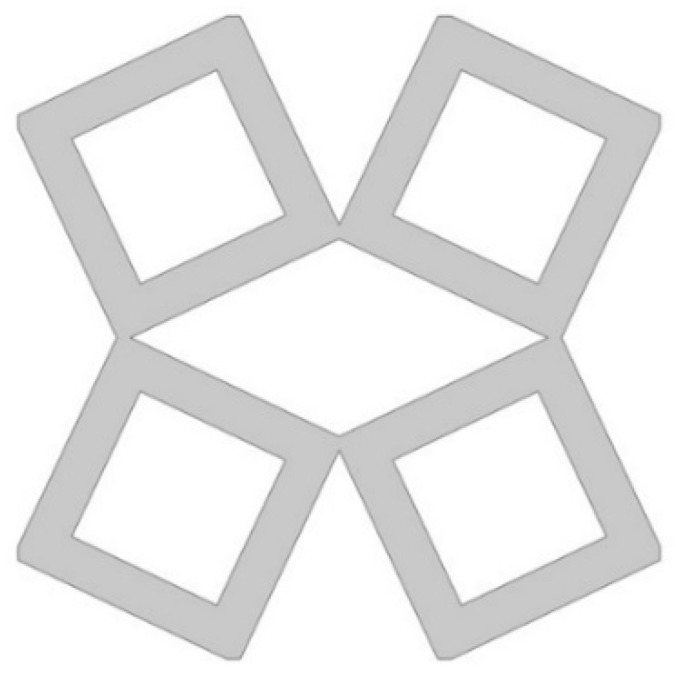
The obtained unit cell for *d*_1_ = 0.65.

**Figure 5 materials-15-08712-f005:**
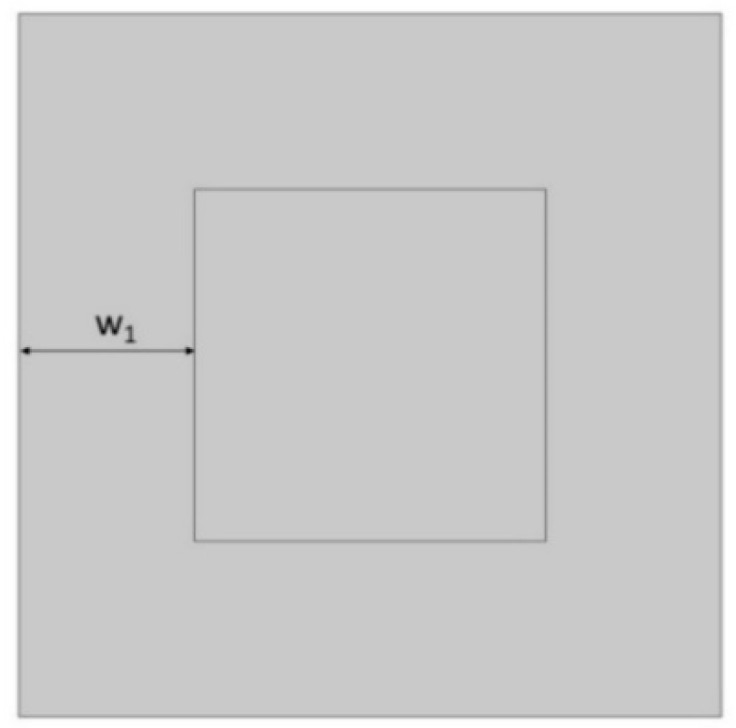
Distance between the outer and inner square in the unit.

**Figure 6 materials-15-08712-f006:**
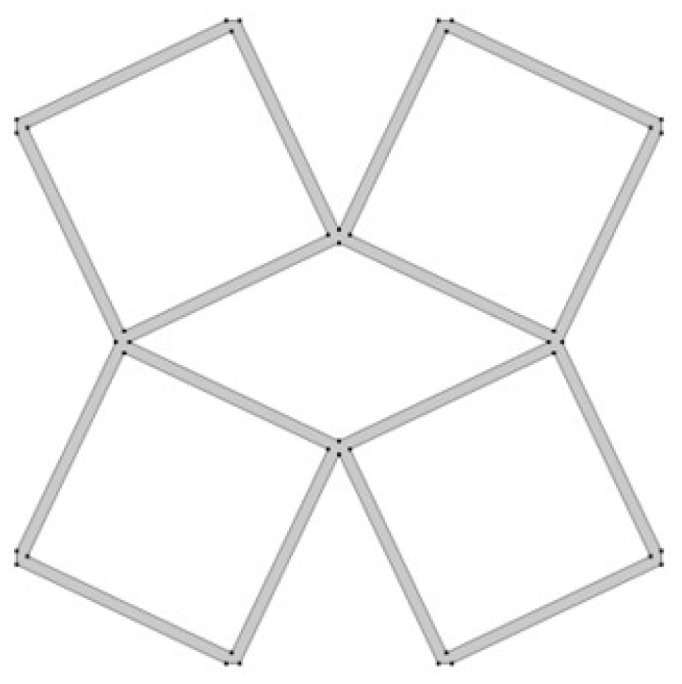
The obtained unit cell for *w*_1_ = 0.15.

**Figure 7 materials-15-08712-f007:**
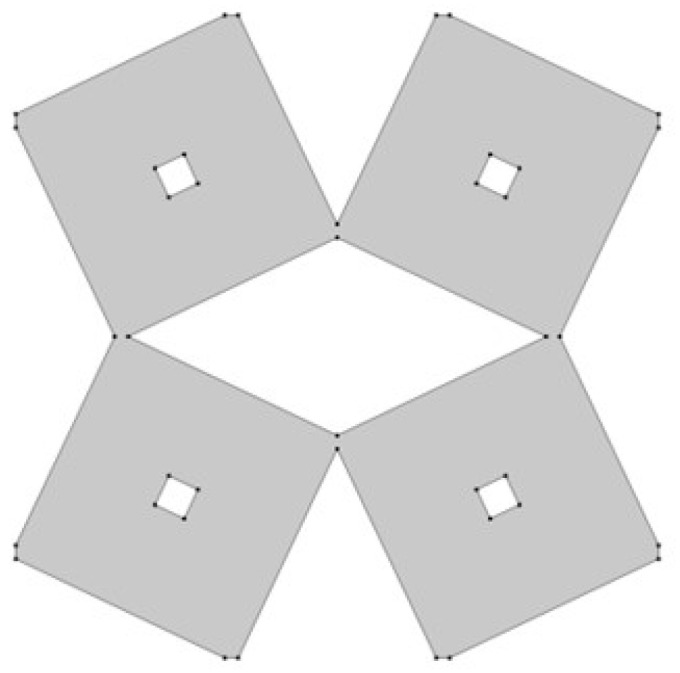
The obtained unit cell for *w*_1_ = 0.45.

**Figure 8 materials-15-08712-f008:**
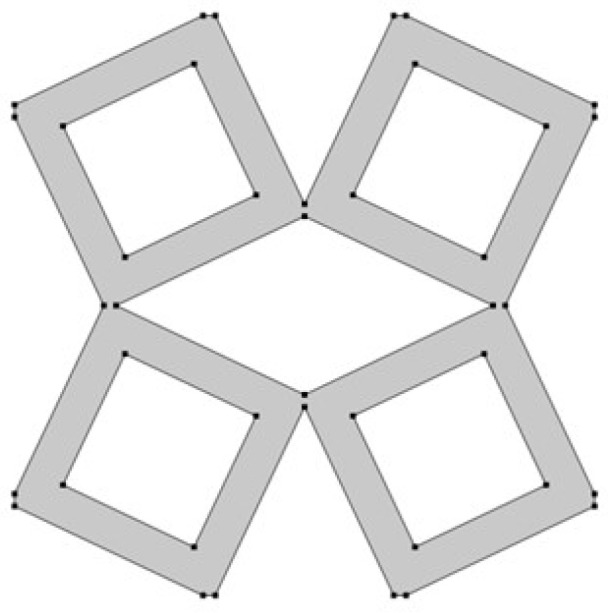
The obtained unit cell for fillet radius = 0.

**Figure 9 materials-15-08712-f009:**
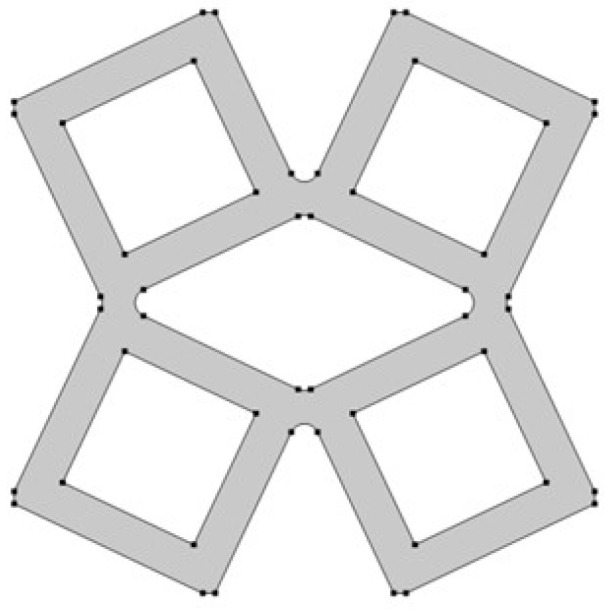
The obtained unit cell for fillet radius = 0.05.

**Figure 10 materials-15-08712-f010:**
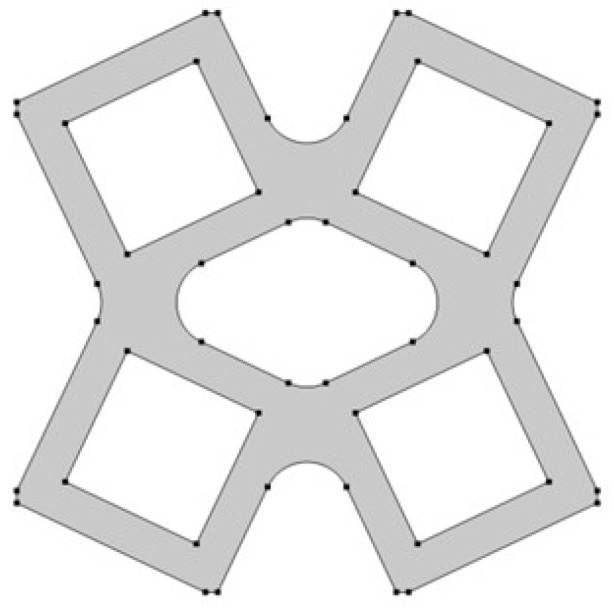
The obtained unit cell for fillet radius = 0.15.

**Figure 11 materials-15-08712-f011:**
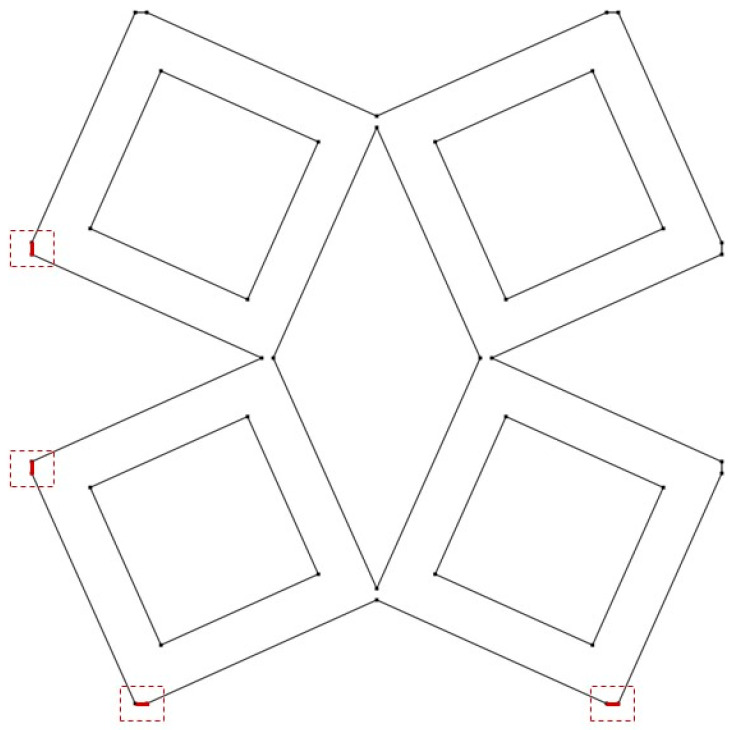
The edges constraint using roller boundary condition.

**Figure 12 materials-15-08712-f012:**
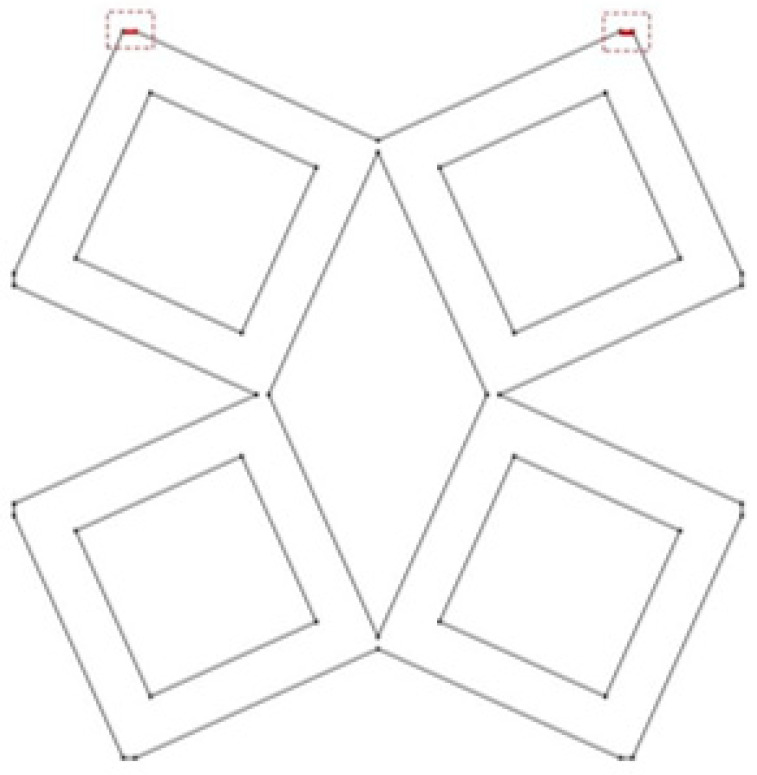
Considered load scenario for evaluation of effective structure properties. Free boundary condition is applied for the remaining boundaries.

**Figure 13 materials-15-08712-f013:**
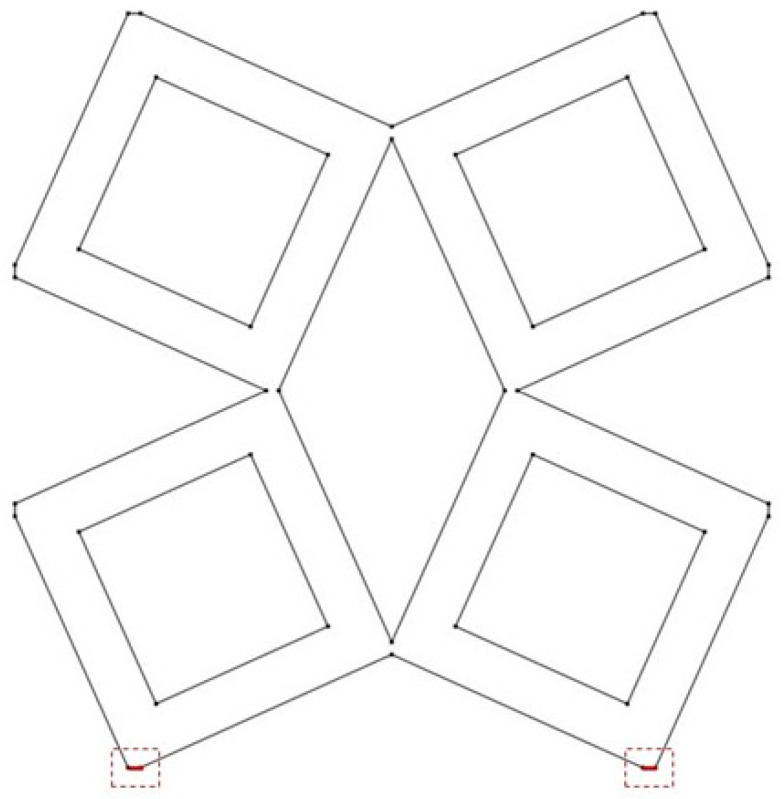
Considered load scenario for evaluation the dynamic characteristic of the structure. Free boundary condition is applied for the remaining boundaries.

**Figure 14 materials-15-08712-f014:**
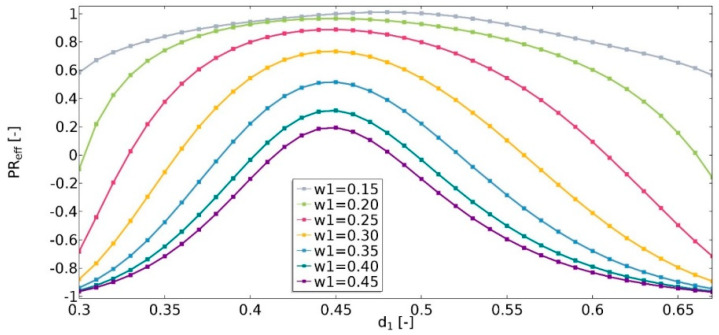
The influence of geometric features on the value of effective Poisson’s ratio.

**Figure 15 materials-15-08712-f015:**
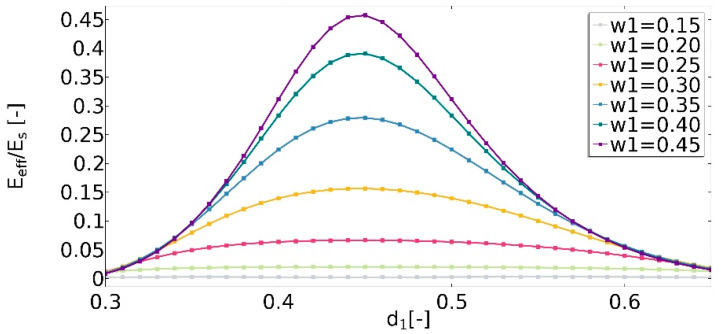
The influence of geometric features on the value of the ratio of the effective Young’s modulus to Young’s modulus of structure.

**Figure 16 materials-15-08712-f016:**
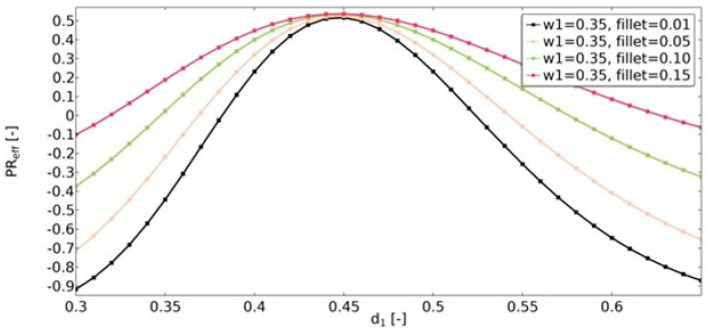
The influence of fillet radius on the value of effective Poisson’s ratio for *w*_1_ = 0.35.

**Figure 17 materials-15-08712-f017:**
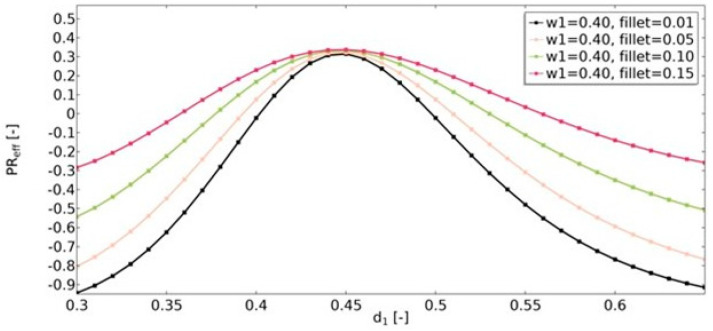
The influence of fillet radius on the value of effective Poisson’s ratio for *w*_1_ = 0.40.

**Figure 18 materials-15-08712-f018:**
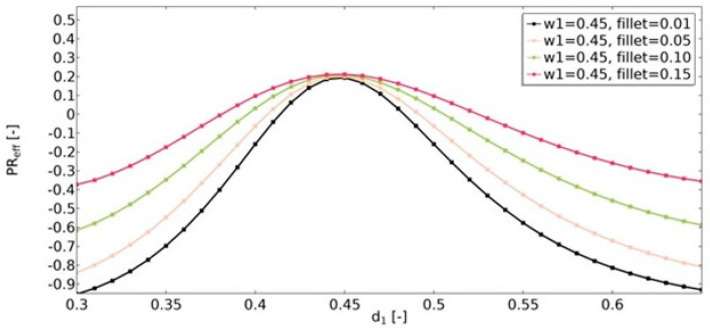
The influence of fillet radius on the value of effective Poisson’s ratio for *w*_1_ = 0.45.

**Figure 19 materials-15-08712-f019:**
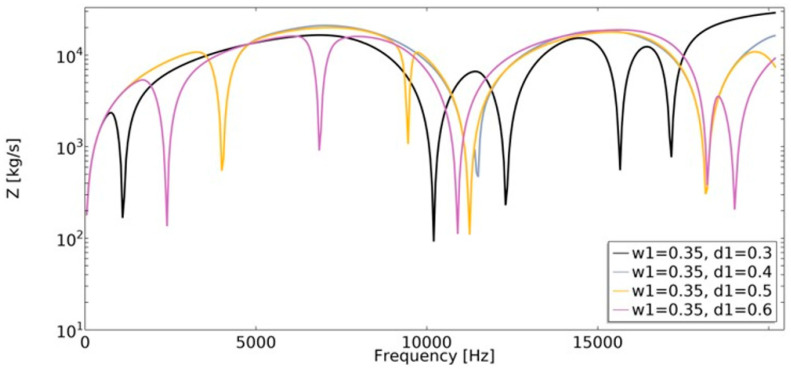
The influence of geometric features on the value of mechanical impedance in the frequency range of 1–20,200 Hz for *w*_1_ = 0.35.

**Figure 20 materials-15-08712-f020:**
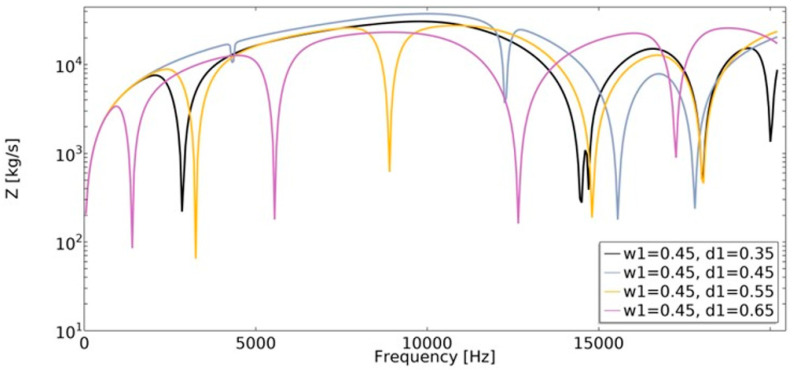
The influence of geometric features on the value of mechanical impedance in the frequency range of 1–20,200 Hz for *w*_1_ = 0.45.

**Figure 21 materials-15-08712-f021:**
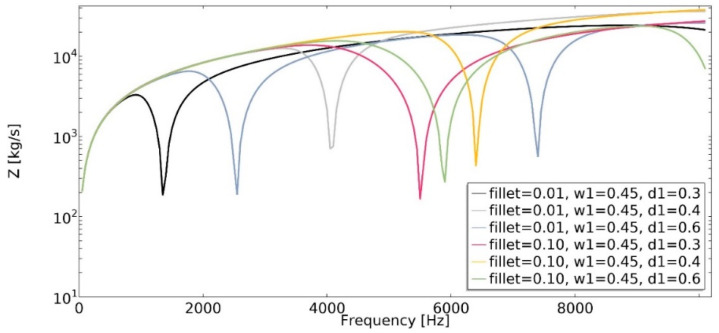
The influence of fillet radius on the value of mechanical impedance in the frequency range of 1–10,100 Hz.

**Figure 22 materials-15-08712-f022:**
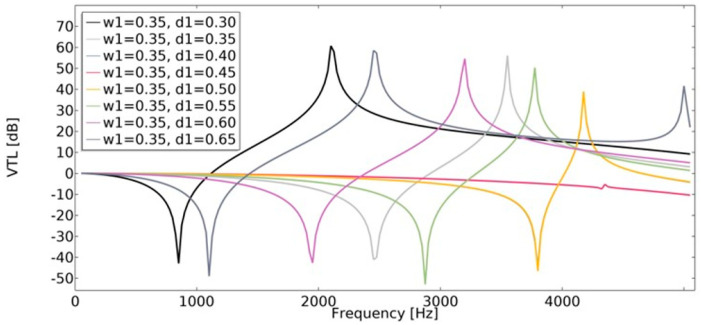
The influence of geometric features on the value of VTL for *w*_1_ = 0.35.

**Figure 23 materials-15-08712-f023:**
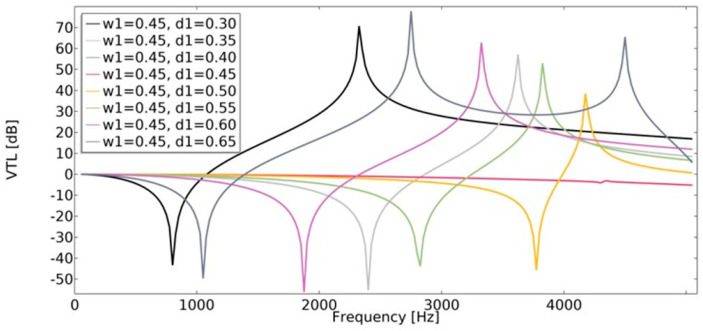
The influence of geometric features on the value of VTL for *w*_1_ = 0.45.

**Figure 24 materials-15-08712-f024:**
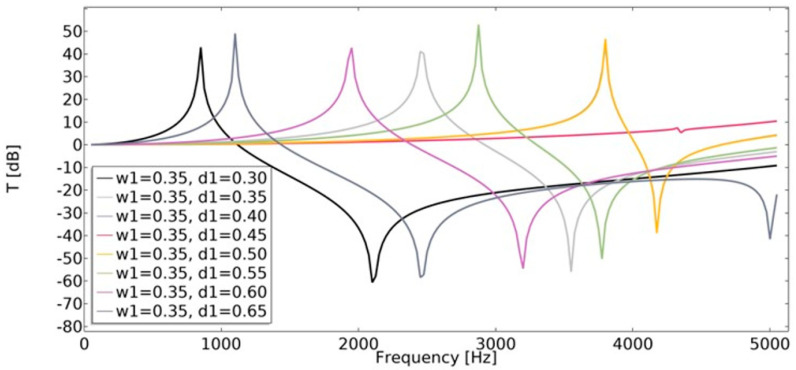
The influence of geometric features on the value of transmissibility in vertical direction for *w*_1_ = 0.35.

**Figure 25 materials-15-08712-f025:**
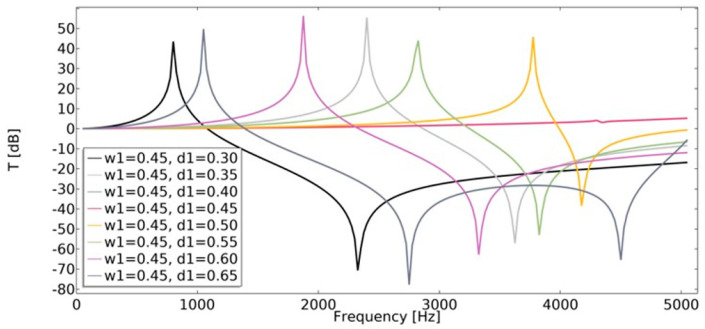
The influence of geometric features on the value of transmissibility in vertical direction for *w*_1_ = 0.45.

**Table 1 materials-15-08712-t001:** Material properties of structural steel.

Property	Value [unit]
Density	$7850\;\left[\mathrm{kg}/m^{3}\right]$
Young’s modulus	200e9 [Pa]
Poisson’s ratio	0.30 [-]
Isotropic loss factor	0.0024 [-]

**Table 2 materials-15-08712-t002:** Ranges of analyzed values of parameters.

Parameter	Range of Parameter Values [-]
*d* _1_	0.30–0.65
*w* _1_	0.15–0.45
fillet	0.00–0.15

## Data Availability

Some or all data, models, or code that support the findings of this study are available from the corresponding author upon reasonable request.

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
