# Peer review of "Numerical Analysis of Dynamic Properties of an Auxetic Structure with Rotating Squares with Holes"

_materials, 2022, doi:10.3390/ma15248712_

Round 1

Reviewer 1 Report

In this paper, a novel structure with rotating squares with holes was investigated. The structure unit cell consists of four units in the shape of a square with cut corners and holes. Such structure represents a kind of modified auxetic structure made of rotating squares with 11 holes or sheets of material. I can see authors have put a lot of efforts. This work, therefore, is recommended to be published after a minor modification.

1.       Literature review, it is necessary to refer to following articles:

(1)      APPLIED MATHEMATICAL MODELLING, 2018, 57: 448-458.

(2)      COMMUNICATIONS IN NONLINEAR SCIENCE AND NUMERICAL SIMULATION, 2018, 56: 380-391.

For better reading, the theories of this model may be to be clearly introduced deeply. 

Please check in detail whether formulas in this manuscript are all in the correct format and explain.

English language and style can be improved and spell check required a little.

It can be published after minor modification.

Author Response

DEAR EDITOR AND REVIEWERS

Thank you for your letter and for the reviewers’ comments concerning our manuscript entitled "Numerical Analysis of Dynamic Properties of Auxetic Structure with Rotating Squares with Holes" (Research Article, No. materials-2051007).

Those comments are all valuable and helpful for revising and improving our paper. We have studied all comments carefully and have made the conscientious correction. All revised portions of the manuscript are marked in other colors (red and blue). Additionally, all added sentences and most of the changes are denoted with a yellow background.

We present our point-by-point response to the comments raised by the reviewers in the attached pdf file (materials-2051007-response-221130.pdf).

We would like also to thank you for allowing us to resubmit a revised copy of the manuscript.

Thank you,

Tomasz Strek (on behalf of all authors)

Reviewer 2 Report

The Authors address the computational study of two-dimensional auxetic structures in plane stress conditions. The analysis aims at characterizing the dependence of the Poisson ratio (in the range [-1, +1]) on several properties of the examined structures. The topic is of interest in several fields of applied sciences, as illustrated at Page 3. However, despite the large amount of produced simulation data, the manuscript is affected by limitations that can be summarized as follows:

1. Lack of connection between the examined "unit cells" and their possible use as building elements of a realistic structure. This limitation makes the description of the test cases more academic than practical.

2. Lack of comment on the choice of the basic size of the examined structures, i.e., Lx=Ly=0.1m. It is not clear whether a different scaling of unit cell size may have an impact on the simulation results. Likely, this should not be the case, because the Authors are assuming a linear constitutive stress-to-strain relation. However, it is not clear what the results might be if a different (nonlinear) stress-to-strain law is assumed.

3. In Table 1, page 11, data for structural steel are considered. Steel is (evidently) not an auxetic material. Is this choice made as a reference for further comparison? If so, this must be definitely clarified.

4. As a general comment, despite the extensive analysis as a function of geometrical, mechanical and electromagnetic parameters is certainly of interest, it is also to be said that the Authors merely produce a list of results, which are not connected to each other in a unified framework. It would be very important that the Authors try to provide such a framework, in terms of relevance of simulation tests in realistic applications, in the revised version of the manuscript.

5. The quality of the English language is often very poor and considerable editing must be performed before the manuscript can be accepted for publication. Examples;

a. "the lower W, the higher Y", instead of the phrases used in the text.

b. filet -----> fillet everywhere in the text.

c. proofed -----> proved everywhere in the text.

d. many times, the use of "the" is completely missing.

Examples:

Page 4, line 173: According to aforementioned equations ------> According to the aforementioned equations.

Page 5, line 204: is direction of applied load -----> is the direction of the applied law.

Author Response

(The authors gave the same response as above.)

Round 2

Reviewer 2 Report

The authors have responded to all the observations and remarks made in the first review in a sufficiently satisfactory manner. Therefore, the revised manuscript can be considered adequate for publication.